# A 3D-Printed Capacitive Smart Insole for Plantar Pressure Monitoring

**DOI:** 10.3390/s22249725

**Published:** 2022-12-12

**Authors:** Anastasios G. Samarentsis, Georgios Makris, Sofia Spinthaki, Georgios Christodoulakis, Manolis Tsiknakis, Alexandros K. Pantazis

**Affiliations:** 1Institute of Electronic Structure and Laser, Foundation for Research and Technology Hellas, 70013 Heraklion, Greece; 2Department of Physics, University of Crete, 70013 Heraklion, Greece; 3Department of Electrical and Computer Engineering, Hellenic Mediterranean University, 71410 Heraklion, Greece

**Keywords:** gait analysis, wearable sensors, 3D printing, capacitive pressure sensors, smart insole, real-time plantar pressure monitoring

## Abstract

Gait analysis refers to the systematic study of human locomotion and finds numerous applications in the fields of clinical monitoring, rehabilitation, sports science and robotics. Wearable sensors for real-time gait monitoring have emerged as an attractive alternative to the traditional clinical-based techniques, owing to their low cost and portability. In addition, 3D printing technology has recently drawn increased interest for the manufacturing of sensors, considering the advantages of diminished fabrication cost and time. In this study, we report the development of a 3D-printed capacitive smart insole for the measurement of plantar pressure. Initially, a novel 3D-printed capacitive pressure sensor was fabricated and its sensing performance was evaluated. The sensor exhibited a sensitivity of $1.19\;\mathrm{MP}a^{-1}$, a wide working pressure range (<872.4 kPa), excellent stability and durability (at least 2.280 cycles), great linearity ($R^{2}=0.993$), fast response/recovery time (142–160 ms), low hysteresis (DH<10%) and the ability to support a broad spectrum of gait speeds (30–70 steps/min). Subsequently, 16 pressure sensors were integrated into a 3D-printed smart insole that was successfully applied for dynamic plantar pressure mapping and proven able to distinguish the various gait phases. We consider that the smart insole presented here is a simple, easy to manufacture and cost-effective solution with the potential for real-world applications.

## 1. Introduction

Human gait, as natural and as simple as it seems, is the product of a complicated and cooperative process involving the brain, spinal cord, nerves, muscles, bones and joints. Gait is a typical activity for healthy human beings but also characteristic of a person’s style and quality of life. Proper gait functionality is essential for maintaining an abundant lifestyle, healthier and happier, while any deviation from the standard can drastically affect everyday experiences.

The study of human walking, namely gait analysis, can be used as a valuable diagnostic tool to distinguish between normal and pathological gait. Abnormal gait patterns are related to the pathology of the human locomotor system, which can be caused by various pathological conditions including neurodegenerative, musculoskeletal or other peripheral disorders [1]. For example, apraxic gait is characterized by the deterioration of neurons, loss of locomotion control and eventually inability of proper movement as a result of diseases such as Parkinson’s [2], Alzheimer’s [3] or cerebral palsy (CP) [4], etc. Limited mobility and dysfunctional gait, e.g., antalgic patterns, may be the effect of chronic musculoskeletal pain arising from age-related conditions such as knee osteoarthritis [5], osteoporotic hip fractures [6], etc. In this respect, a quantitative gait analysis at a specific moment, or through continuous monitoring and re-evaluation over time, can support early diagnosis of diseases, credible clinical decisions, optimization of treatment protocols and assessment of patient outcomes [7]. Apart from clinical applications, the reliable analysis of human gait characteristics is one of the main interests in the fields of sports science [8], rehabilitation [9], security monitoring [10] and robotics [11].

The traditional/old-fashioned approach to human gait analysis is based on the observation of a patient’s gait performed in a clinic by experienced specialists, aided with patient self-reporting. However, this method is considered unreliable and oversimplified, i.e., the evaluation is subjective and lacks of quantitativeness [1]. The progress of new technologies has given rise to more delicate, advanced techniques which allow for an accurate, objective and quantitative measurement of gait parameters [12]. The standard method for gait analysis, adopted by the majority of specialized clinics, is a vision-based motion analysis captured by digital video cameras and thoroughly analyzed through image processing techniques [13]. Alternatively or in combination, force platforms can be used to extract gait information; they are located on the floor and utilize a pressure mattress to measure ground reaction forces (GRFs) during walking [14]. Nonetheless, the aforementioned methodologies are relatively expensive, time-consuming and require expertise [15]. Moreover, the examination procedure is strictly performed to controlled environmental settings, where the patient is aware of his/her movements and usually with markers attached to the body; hence, the results do not reflect real-world activities [15].

In the last decade, wearable insole-based sensor systems have shown great potential as gait analysis tools, intending to overcome the aforementioned limitations of stationary force platforms [16,17,18]. These wearable technologies allow for continuous gait monitoring in both indoor and outdoor environments on a daily basis without any constraints on an individual’s natural gait during measurements. Most often, they are used to measure the plantar pressure distribution, despite the fact that other gate features can also be obtained, including the center-of-pressure (CoP), step count, duration of gait cycle, swing and stance duration [19]. Insole-based plantar pressure sensors produce an electrical signal upon pressure loading during human gait. Based on their working mechanism, they can be divided into three major, widely used types: piezoelectric [20,21,22], piezoresistive [23,24,25,26] and capacitive [27,28,29,30]. Each sensing mechanism exhibits its own merits and limitations [16,18], none of them has a clear edge and it is rather hard to develop a sensor device with ideal characteristics [31]. The basic parameters to determine the performance of plantar pressure detection sensors are sensitivity, linearity, range of detection, response time, hysteresis, stability, durability and interferences from external sources such as temperature, humidity or electromagnetic interference [32]. Apart from sensor performance characteristics, multiple factors have to be considered prior to the development of a new insole sensing system, including sensing elements layout, electronics circuit design, signal processing algorithms, energy consumption, manufacturing cost and user comfort [7,16,18].

Capacitive pressure sensors have been established as a compelling candidate for insole-based plantar pressure monitoring due to their simple structure, low power consumption, good reliability, repeatability and dynamic performance [16,17,33]. Nonetheless, their performance can be affected by humidity, temperature and electromagnetic interference [16]. Many attempts have been made over the years to integrate capacitive pressure sensors into smart insoles. In order to increase plantar pressure sensitivity, capacitive sensors with porous polydimethylsiloxane (PDMS) as the dielectric material have been repeatedly presented in the literature [34,35,36]. In another approach, Zhang et al. developed a low-cost, capacitive-based plantar pressure sensor composed entirely of fabric [37]. Tao et al. established a real-time pressure mapping smart insole system with a rubber dielectric layer and studied various motions and postures [29]. Aqueveque et al. presented a gait segmentation method using a custom-made capacitive insole [28]. Sorrentino et al. developed a capacitive insole prototype with temperature compensation and high spatial resolution [27]. Recently, De Guzman et al. created a low-cost, capacitive insole for plantar pressure measurement as a possible alternative to more expensive systems [30]. However, the insoles presented in the aforementioned studies are characterized by complex and expensive fabrication processes or inadequacy for cost-effective mass production. In addition, a few capacitive-based insole products have become commercially available in the last two decades. For example, the Pedar^®^ (Novel GmbH, Munich, Germany) pressure distribution measuring system is a widely used and well-tested in-shoe device based on 99 capacitive sensors [38,39,40]. Another product worth mentioning is Moticon’s (Moticon GmbH, Munich, Germany) OpenGo^®^ sensor insole system which consists of 16 capacitive pressure sensors and is capable of measuring plantar pressure distribution and acceleration in three dimensions [41,42,43]. PODOSmart^®^ (Digitsole SAS, Nancy, France) insoles have also been validated for normal walking measurements using a stereophotogrammetry-based system [44]. Despite the above solutions having been proven in terms of accuracy and repeatability, they are characterized by a relatively high cost.

In recent years, the application of three-dimensional (3D) printing technology for the manufacturing of sensors has attracted a significant amount of research interest both in industry and academia [45]. When compared to traditional semiconductor processing techniques, 3D printing offers several advantages, including lower fabrication costs, reduced manufacturing time, lesser number of processing steps, a range of different materials and prototypes that can be easily customized according to application [46]. Various types of 3D-printed sensors such as force, acoustic and ultrasonic, optical and electromagnetic have been developed for engineering [45,47] and biomedical applications [46,48,49]. Several 3D-printed capacitive-based, flexible strain and tactile sensors can be found in the literature [50,51], but to the best of our knowledge, a 3D-printed capacitive smart insole for plantar pressure monitoring has not been developed up to now.

In this study, we present the development of a 3D-printed smart insole suitable for plantar pressure monitoring. The insole incorporates 3D-printed pressure sensors based on the capacitive sensing principle. At first, the capacitive sensors were tested under dynamic loading conditions to assess their performance by measuring their principal characteristics. These include sensitivity, linearity, pressure detection range, durability, stability, hysteresis and response/recovery time. The 3D-printed sensors were integrated into a 3D-printed insole, demonstrating the potential use of the system for recording gait-related data.

## 2. Materials and Methods

### 2.1. A 3D-Printed Capacitive Pressure Sensors

Capacitive pressure sensors were fabricated using 3D printing technology, employing the Fused Filament Fabrication (FFF) method. Specifically, 3D printing was performed with the TENLOG TL-D3 Pro Dual Extruder 3D Printer (TENLOG 3D solutions, Shenzhen, China). The sensors were designed via the Fusion 360 CAD software (Autodesk, San Rafael, CA, USA), while the Ultimaker Cura 4.6.2 software (Ultimaker B.V., Utrecht, The Netherlands) was used for the slicing of the object into layers.

Capacitive pressure sensors consist of two conductive plates separated by a dielectric material. As it is shown in Figure 1, the in between section of the two plates consists of two dielectric materials, in separate levels: First, an air cavity with a thickness of 1.0 mm and then a 0.2 mm solid flexible material (Filaflex 70A). This architecture was chosen as it offers greater sensitivity than the classic plate–dielectric–plate structure [52]. We should mention here, that prior to the presented architecture, we had already tested various versions of the classic capacitive structure (data not shown), where the dielectric was either solid or had a grid pattern (20% to 100% infill). However, in this case the system demonstrated a poor signal response/sensitivity. In general, the pressure sensitivity of a capacitive pressure sensor $S_{c}$ can be expresses as [34]:(1)$$S_{c}=\frac{\left(\Delta C/C_{0}\right)}{\Delta P}$$
where ΔC is the change in capacitance, $C_{0}$ is the initial capacitance without the application of pressure and ΔP is the pressure change.

The optimized model of the capacitive pressure sensor incorporates two conductive plates made of Protopasta CDP11705 composite conductive PLA (Protoplant, Vancouver, Canada) while the middle dielectric layer is made of Filaflex 70A TPU (Recreus Industries S.L., Elda, Spain) (Figure 1). The overall thickness of the sensor is t=2 mm and its diameter d=14.5 mm with layers of the following thicknesses: bottom electrode— 0.3 mm, top electrode—0.5 mm and dielectric in total—1.2 mm. The settings for the 3D printing of the different materials used for the fabrication of the capacitive pressure sensors are summarized in Table 1.

The capacitance for two dielectric layers in series can be found easily from basic electrodynamics equations [53] and expressed as:(2)$$C=\frac{\epsilon_{0}\times A}{\frac{d_{1}}{k_{1}}+\frac{d_{2}}{k_{2}}}$$
where $\epsilon_{0}$ is the dielectric constant of free space, A is the area of the plate, $d_{1},\;d_{2}$ are the thicknesses of the two dielectric materials and $k_{1},\;k_{2}$ are the relative permittivity of the two dielectrics, respectively.

When pressure is applied on the capacitance sensor, the two plates (top and bottom) come closer. Thus, the total distance ($d_{1}+d_{2}$) between the two electrodes is reduced on the system and the capacitance changes (Equation (2)).

On both plates, we fix the cables with an adhesive conductive glue and on top of the glue we place a small piece of insulation tape as a protective layer.

### 2.2. Experimental Set-Up for Dynamic Measurements

An in-house experimental set-up was designed and manufactured to facilitate dynamic measurements by simulating a pressure point. This is performed through an aluminum shaft (piston) that moves up and down. The shaft moves through a flanged linear ball bearing (8 mm), which is fixed on the structure (wooden crate). The sensor is located on the bottom side of the structure, just below of the shaft’s tip (area of the tip is equal to the area of sensor). On the top part of the shaft, the crate supports the installment of a pulley system that enables the movement of the shaft on the vertical axis (Figure 2). Additionally, the pulley system is easily disengaged from the shaft in order to place on it 0.5 kg disk weights. In this direction, the pulley system transmits the motion to the piston through a stepper motor (Nema 23 57BYGH115 from Wantai motors), which is located on the side of the crate, and is connected to a motor driver (TB6600), both controlled by an Arduino Nano system (Arduino LLC, Boston, MA, USA). This way, the shaft applies periodic force on the sensor. Two potentiometers control the speed and the elevation of the shaft, which are modified in real time. The gait speed is displayed on a microdisplay (1.3″ OLED Display Module from Waveshare) as steps per minute per foot. A second Arduino Nano board is used for the system’s output acquisition and data are recorded using a code written with the open-source Arduino software language (IDE). The total time of the measurement and the sampling period are defined by the user. The sensor is fixed at this position with adhesive tape (3M™ Acrylic Adhesive 300MP) before the experiment initiation. The set-up makes it possible to measure the sensor’s response as a function of different mass loading conditions ranging from 0.7 (weight of shaft) to 14.7 kg (which in pressure units is 41.5 kPa to 872.4 kPa). The environmental conditions of the sensor tests were 20 °C and 55–60% humidity.

### 2.3. A 3D-Printed Smart Insole

The fabricated 3D-printed capacitive pressure sensors were used for the design and development of a 3D-printed smart insole. The architecture, circuit connection and images of the 3D-printed capacitive smart insoles are presented in Figure 3. The upper and lower layer of the insole are made of Filaflex 82A (Recreus Industries S.L., Elda, Spain) and the printing parameters are the same as those presented in Table 1. Each insole incorporates 16 capacitive pressure sensors which are fixed to the lower flexible case and adequately electrically connected. The sensors are distributed in the insole as depicted in Figure 3A. The total thickness of the insole is 3.8 mm, with a length up to 25 cm and width up to 8.5 cm, corresponding to a No.39 EU size shoes. The two layers (upper and lower) are glued together at the perimeter of the insole.

### 2.4. Smart-Insole Graphical User Interface (GUI) Implementation

The GUI presented in the current investigation is a Java-based desktop application developed in IntelliJ IDEA [54]. It makes use of a “lightweight” GUI Framework, called “Swing”, which contains a set of classes to provide powerful and flexible GUI components [55]. The application communicates with the insole, and particularly with the electronics module, with the use of the *Serial Communication* protocol, through a USB cable, or, alternatively, a wireless Bluetooth module working as a serial (Rx/Tx) pipe. It receives and converts in real-time the sensor readings, acquired from the A/D converter of the electronics module, into pressure values, and subsequently, it presents in a graphical manner the pressure variations of each individual sensor and the center of pressure (CoP) in total too. In the implementation of the individual pressure variations, the insole, scaled into the monitor resolution, is divided into 227 grids as shown in the middle of Figure 4. Then, a group of nearby grids is assigned to every sensor, which, depending the current pressure value, change color accordingly, with yellow indicating the lowest and red the highest pressure. Next to the grid implementation, the individual pressure values may also be presented in a chart-type implementation as shown at the left and the right side of Figure 4.

## 3. Results and Discussion

### 3.1. Sensor Evaluation

In order to evaluate the sensing performance of the 3D-printed capacitive pressure sensors, we conducted a series of experiments that enabled to assess their basic characteristics. The real-time sensor responses under dynamic pressure loading were used to obtain an estimation of their sensitivity, both in and out of the insole. In addition, we investigated other key parameters correlated to sensor performance including linearity, limit of detection (LoD), detection range, effect of loading frequency, stability, durability, repeatability, hysteresis, response and recovery time. In this manner, their potential for practical applications such as plantar pressure monitoring was verified.

#### 3.1.1. Real-Time Measurements

The real-time capacitance changes of a typical 3D-printed capacitive pressure sensor as a function of varying external pressure are demonstrated in Figure 5. The dynamic experiments were performed under a loading frequency that was kept constant at 40 steps/min. The applied pressure ranged from 41.5 kPa to 872.4 kPa with an incremental increase of 59.35 kPa, while one-minute recordings were obtained at every pressure. In these graphs, we present the dynamic responses of the sensors at selected pressure values (step=118.7 kPa) under three loading/unloading cycles. As observed, the capacitance response gradually increases with pressure and its value is clearly distinctive among all steps. Moreover, the sensor is characterized by a stable and repeatable capacitance response under every pressure in the range tested. The results indicate that the sensors can efficiently operate at this pressure range without any damage, since in all cases they rapidly recover to their initial baseline value.

#### 3.1.2. Sensitivity

In order to assess the sensitivity of the 3D-printed capacitive pressure sensors, we measured their relative responses versus external pressure. In Figure 6, the relative capacitance change, the average of 16 sensors, is presented under varying external pressures ranging from 41.5 kPa to 872.4 kPa, using a step of 59.35 kPa. In this range, the mean sensor response was proved to be linear with pressure, while the slope of the curve was used to estimate the sensitivity. As shown in Figure 6, the average sensitivity of the capacitive sensors is $1.19\;\pm 0.03{\;\mathrm{MPa}}^{-1}$, showing a great linear trend ($R^{2}=0.993)$ under the pressure range tested up to 872.5 kPa. The LoD is 41.5 kPa, which means that below this point it is not possible to reliably detect any pressure changes. The sensitivity found here is higher than that of other capacitive pressure sensors with similar working pressure ranges reported previously in the literature [34,37]. Our results suggest that the developed capacitive sensors are well-suited for applications with a broad pressure range.

#### 3.1.3. Loading Frequency

Considering the specific application envisaged for the capacitive pressure sensors, i.e., gait analysis, it is important to verify the sensor response at different loading frequencies, which correspond to altering gait speeds. In this direction, Figure 7 shows the capacitance response of a typical 3D-printed capacitive pressure sensor as a function of different loading frequencies under a pressure load of 872.5 kPa. The range of the tested gait speed is between 30 and 70 steps/min**,** simulating slow, normal, fast and dynamic gait. As shown, no dependence of the sensor response was found on the gait speed. This means that the operation of the capacitive sensor remains stable regardless of the gait pace. In this graph, the initial $C_{0}$ and final $C_{f}$ capacitance values are depicted, as well as their difference ΔC.

#### 3.1.4. Durability

In Figure 8, we present the capacitive sensor response during 1 h of continuous measurements (2.280 cycles of loading/unloading) under an applied pressure of 872.4 kPa. The experiment is used as a reliability test in order to assess the durability of the capacitive sensor. The inset figures demonstrate the first and final minute of the measurements. As seen, the response of $C_{0}$ and $C_{f}$ remains the same, confirming the excellent stability and durability of this type of sensor.

#### 3.1.5. Hysteresis

Hysteresis is a critical parameter to be evaluated considering the practical applicability of any pressure sensor. Hysteresis can be defined as the difference in the output signal between consecutive loading and unloading cycles [56]. Negligible or low hysteresis is a desirable property that allows for consistent and accurate measurement of pressure variations over time. Due to the fact that the sensor has to operate under repeated dynamic conditions, potential unrecoverable deformations could introduce significant measurement errors. Figure 9 depicts the measured hysteresis curves, forward and reverse, as obtained from three consecutive linear loading–unloading pressure cycles. According to this graph, two different responses are distinguished with the unloading curve being slightly higher compared to the loading cycle. Nevertheless, the sensor only exhibits low hysteresis, indicating its capability to obtain reliable and accurate pressure signals.

Quantitively, hysteresis can be assessed using the equation for the degree of hysteresis (DH) which is defined as the percentage of the relative difference in the area underneath the loading and unloading curves, calculated by the following equation [56]:
(3)$$DH\left(\%\right)=\frac{A_{Unloading}\;–\;A_{Loading}\;}{A_{Loading}}\times 100$$
where $A_{\mathrm{Loading}}$ and $A_{\mathrm{Unloading}}$ are the areas underneath the curves in Figure 9, corresponding to the loading and unloading cycles, respectively. For our sensor, the DH value was estimated to be 9.8%, implying a relatively good hysteresis level.

#### 3.1.6. Response/Recovery Time

Real-time pressure monitoring requires sensors that exhibit fast response and recovery times. Response time is the interval needed for the sensor output to reach its final value upon a pressure change. Reversely, recovery time is the interval required for a sensor to return to its base value upon pressure release. In this study, the response time was estimated as the time in which the capacitance changes from 10 to 90% of its maximum change upon pressure loading while the recovery time was from 90 to 10% upon pressure unloading [57]. The response/recovery time can be seen in Figure 10, where the sensor response is presented under the loading–unloading of 872.4 kPa. As illustrated, the response/recovery time of our device was found to be in the millisecond range. Specifically, the sensor can detect pressure changes with a response time of 142 ms and a recovery time of 160 ms. This means that the developed sensing device can successfully monitor plantar pressure variations, at least in cases where the stride frequencies are in the range of 30–70 steps/min. The response/recovery time found in this study is similar to that of other capacitive pressure sensors [29,35].

#### 3.1.7. Insole Sensitivity

Since the capacitive pressure sensors developed in this research are meant to be fixed inside an insole, one should consider investigating the effect of the insole and electrical connections on the sensor’s performance. For this purpose, the same sensors were placed inside the designed 3D-printed insole and re-calibration was performed using the same experimental parameters (see Section 3.1.2). The average relative capacitance change ratio $\left(\Delta C/C_{0}\right)$ of sixteen sensors as a function of increasing applied pressure, both outside and inside the insole, is presented in Figure 11. Again, the slope of the curve reveals the average sensitivity of the capacitive pressure sensor. As anticipated, the insole has a noticeable impact on the sensor’s response, i.e., the sensitivity decreases 2.1 times, specifically from 1.19 to $0.55{\;\mathrm{MPa}}^{-1}$. Even so, the linearity is still high $\left(R^{2}\sim 0.989\right)$, and the working pressure range remains the same while the average error level drops to 10%. The insole sensitivity may be further improved by reducing the thickness of the insole’s upper/lower cases and minimizing the electrical noise by upgrading/optimizing the external electrical circuit and hardware.

### 3.2. Plantar Pressure Mapping

To demonstrate the relevance of the developed 3D-printed capacitive pressure sensors for practical applications, we placed the sensors inside a 3D-printed smart insole, which was subsequently used for real-time plantar pressure measurements and gait analysis. The smart insole was designed to incorporate 16 pressure sensors, distributed appropriately across the main pressure areas of the foot during walking. Specifically, the sensors were fixed at four areas: phalanges (T1, T2, T3, T4), metatarsals (ME1, ME2, ME3, ME4, ME5), plantar arch (M1, M2, M3, M4) and calcaneus (H1, H2, H3). The arrangement of the sensing elements inside the smart insole is depicted in Figure 12G.

The smart insole system was fixed at the right foot of a volunteer subject and real-time recordings were obtained during repeated gait cycles. The subject recruited for these experiments was a healthy female, 27 years old, 59 kg in weight and 1.69 cm in height with a European 39 shoe size. No discomfort was reported from the volunteer while wearing the insole. The plantar pressure distribution during a dynamic gait cycle is presented in Figure 12. The heatmaps shown in this graph correspond to the main stance phases of a gait cycle including stance, heel strike, foot flat, midstance, heel off and toe off. The plantar pressure mapping is based on a color scale of ten regions (0–450 kPa), ranging from yellow (low pressure) to red (high pressure). It can be clearly seen that the developed smart insole system was able to assess dynamic pressure variations and successfully determine gait events during a gait cycle. The plantar pressure distribution changes depend on the specific gait phase. In the stance and foot flat phases, a relatively even distribution of the pressure across the foot is observed. In the heel strike stage, ground contact is initiated, which results in the activation of the heel sensors that reach their peak value while the rest of the sensor remains inactive. The midstance posture is characterized by the relocation of the pressure loading mainly on midfoot and forefoot as recorded by the metatarsal and toe sensors. During the heel off phase, the pressure is concentrated on the metatarsals zone and the big toe. Finally, as displayed in the graph, at heel off all sensors return close to zero pressure values, despite the fact that some signal drift or delay is observed. In Figure 12, the trajectory of the center-of-pressure (CoP) is also shown (blue marker) during the gait cycle and its transition through the subphases is revealed. As expected, the CoP is initially located near the heel at the time of heel strike and gradually moves forward to the forefoot near the toes at the heel off stage, indicating a regular variation of the CoP. It should be noted that a proper estimation of the CoP, its trajectory and velocity, is an important parameter to evaluate the balance ability of an individual.

The real-time plantar pressure signals, as collected from the sixteen sensors of the insole during the normal gait cycle, are displayed in Figure 13. All heel sensors are activated at the first three phases, namely stance, heel strike and foot flat with the highest amplitude of pressure measured at sensor H1 during heel strike (~295 kPa). The hallux area, represented here by the sensing element T4, experienced a pressure value of ~127 kPa in the transition between midstance and heel off. At the same time point, the highest pressure value in the phalanges zone was reported at sensor T1 (~370 kPa). The metatarsals area is for the most part loaded between the end of heel strike and the start of heel off. The maximum pressure recorded in this zone was of value ~380 kPa at sensor ME5, which is located lateral to the first metatarsal. The plantar arch received the lowest pressure and it was activated primarily at the foot flat stage with pressures remaining below 150 kPa. Sensors T2 and T3 of the middle toes area did not bear any pressure throughout the gait cycle. The results presented above confirm the applicability of the proposed 3D-printed smart insole system for plantar pressure monitoring in real time with adequate accuracy and time resolution. However, it should be noted that the results depend firmly on the architecture of the sensors, their position within the system (including how close they are packed together), on the manufacturing materials and on the general design of the insole and how it may absorb the pressure during a gait analysis.

## 4. Conclusions

In this work, a 3D-printed smart insole system was established, aiming to monitor the plantar pressure distribution during human gait. The insole system consists of 16 3D-printed pressure sensors employing capacitive sensing. The capacitive sensors were tested under dynamic loading conditions using a custom-built experimental set-up that was developed for gait simulation experiments. The experimental set-up allows to evaluate the response of the sensors under repeated loading/unloading cycles and by varying the loading frequency and amount of pressure. As a result, it was possible to assess the performance of the sensors in terms of real-time dynamic response, sensitivity, linearity, pressure detection range, stability, durability, hysteresis, response/recovery time and gait speed. It was confirmed that the sensors exhibit a very positive performance for their specific application, i.e., real-time human gait monitoring. Using the smart insole and dedicated software, it was possible to obtain plantar pressure maps at different gait phases, to demonstrate the capabilities of our system for gait signal analysis. Moreover, this smart insole system is characterized by a fast and low-cost fabrication process. Depending on the 3D printing systems, one can have a pair of insoles ready within a day. According to Table 2, the manufacturing costs of the 3D-printing parts per sole is a little bit less than EUR 6. The total cost of the insole pair, including the external recording unit, reaches EUR 60. Additionally, the process offers the ability to easily redesign an insole that will specifically meet the requirements of a process and the specificity of a person’s/patient’s foot. This makes the use of the aforementioned smart insole system perfect for personalized applications.

## Figures and Tables

**Figure 1 sensors-22-09725-f001:**
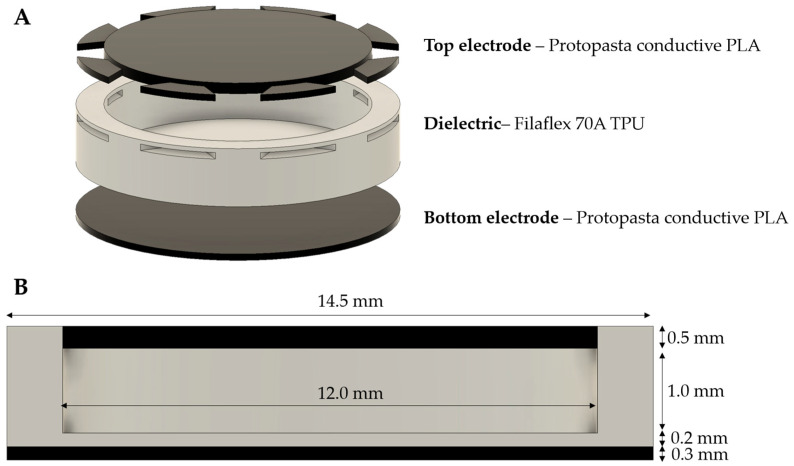
(**A**) Layers of the structure and (**B**) cross-section of the 3D-printed capacitive pressure sensor.

**Figure 2 sensors-22-09725-f002:**
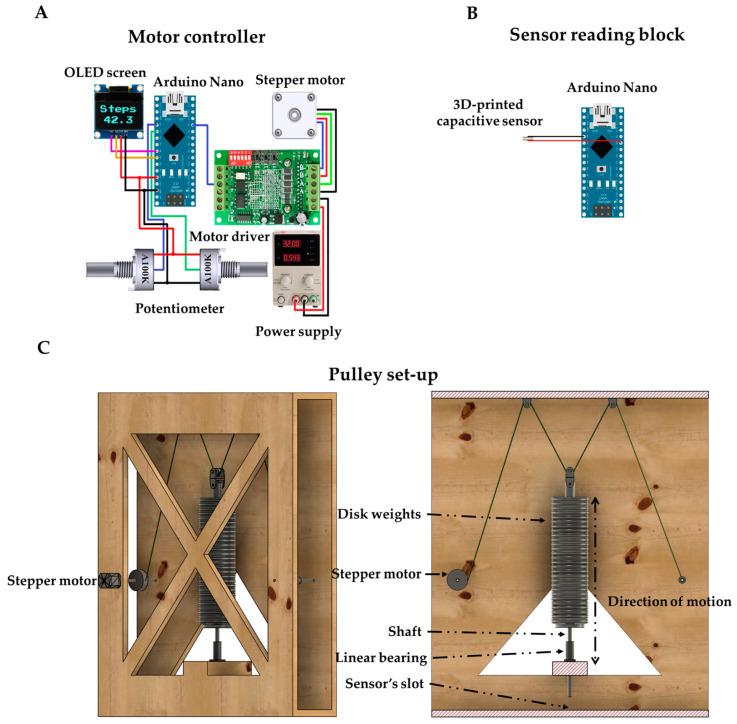
Experimental set-up for dynamic measurements; (**A**) motor control circuit, (**B**) capacitive sensor reading block and (**c**) pulley set-up system.

**Figure 3 sensors-22-09725-f003:**
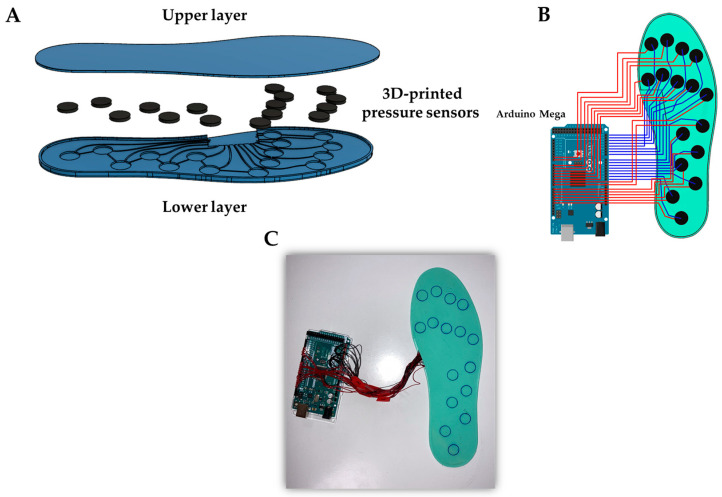
Schematic illustration of 3D-printed smart insole; (**A**) general structure, (**B**) electrical circuit and (**C**) image of the capacitive 3D-printed smart insole.

**Figure 4 sensors-22-09725-f004:**
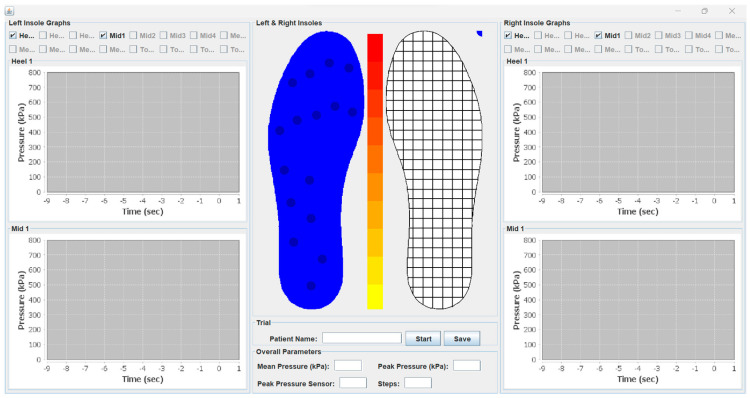
Smart insole graphical user interface (GUI) implementation.

**Figure 5 sensors-22-09725-f005:**
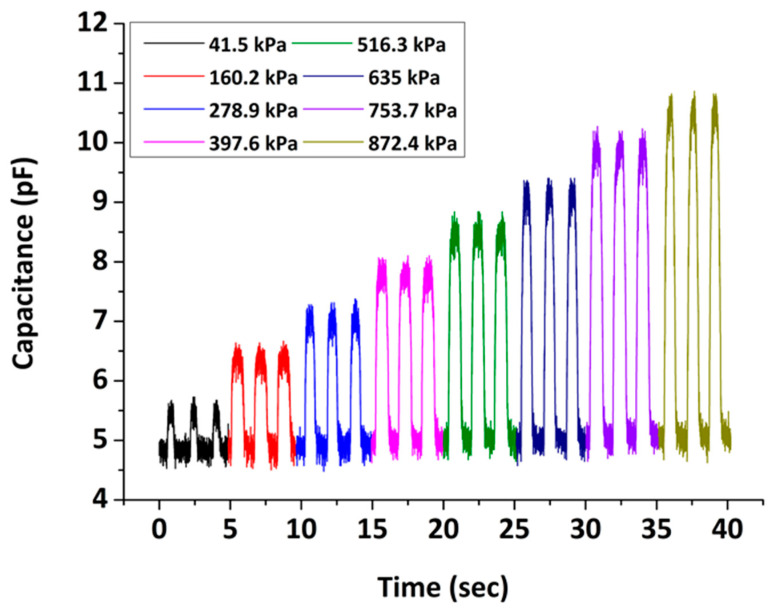
Real-time capacitance response of a typical 3D-printed capacitive pressure sensor at different external pressures under three loading/unloading cycles.

**Figure 6 sensors-22-09725-f006:**
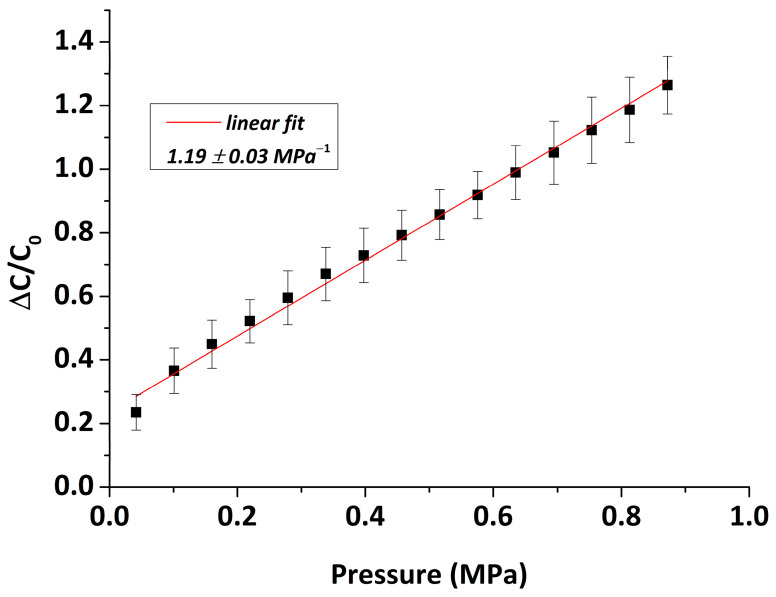
Average relative capacitance variation of 16 sensors as a function of the external pressure applied (0.04–0.87 MPa); the red line represents a linear fit of slope $1.19\;\pm \;0.03{\;\mathrm{MPa}}^{-1}$.

**Figure 7 sensors-22-09725-f007:**
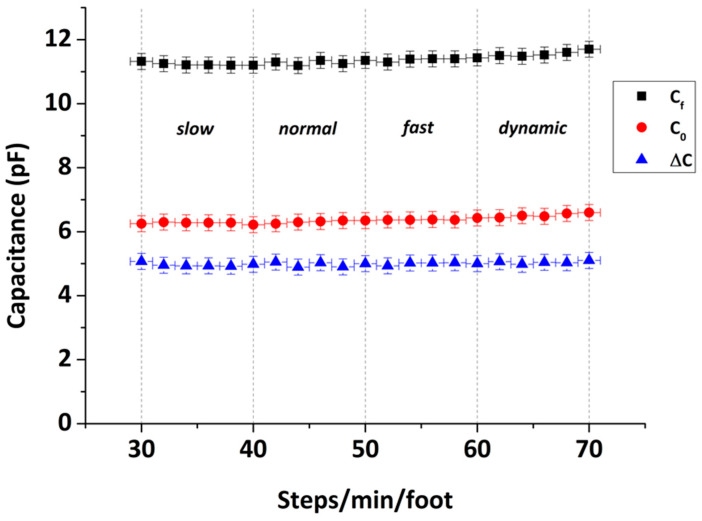
Capacitance response of a typical 3D-printed capacitive pressure sensor at different gait speeds (30–70 steps/min) under an applied pressure of 872.4 kPa; $C_{0}—$ initial value, $C_{f}—$ final value and $\Delta C=C_{f}-C_{0}$.

**Figure 8 sensors-22-09725-f008:**
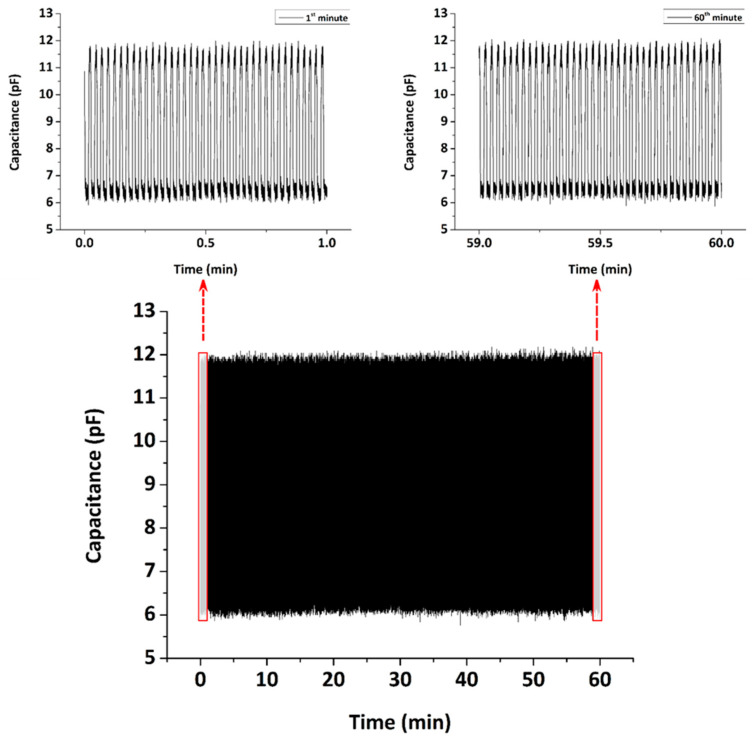
Capacitance response of a typical 3D-printed capacitive pressure sensor during 1 h of continuous measurements at 40 steps/min gait speed under an applied pressure of 872.4 kPa.

**Figure 9 sensors-22-09725-f009:**
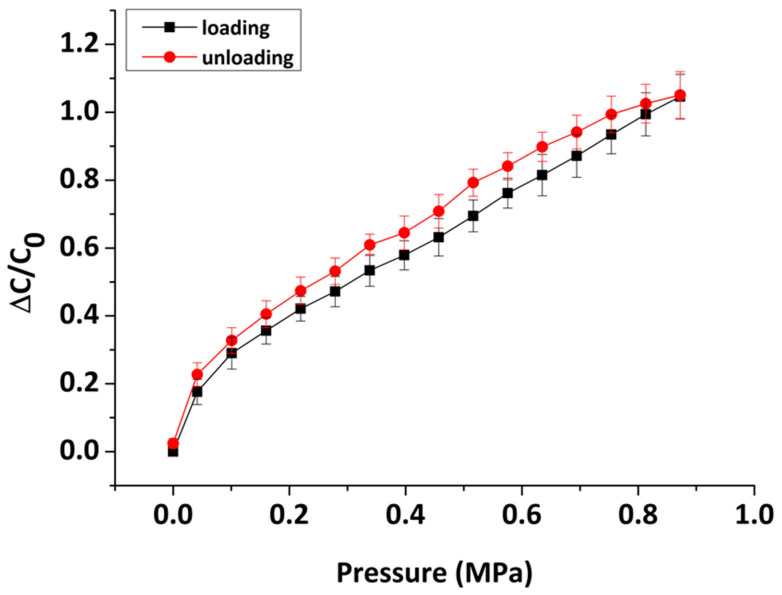
Hysteresis curves of the sensor obtained from three consecutive linear loading–unloading cycles of pressure.

**Figure 10 sensors-22-09725-f010:**
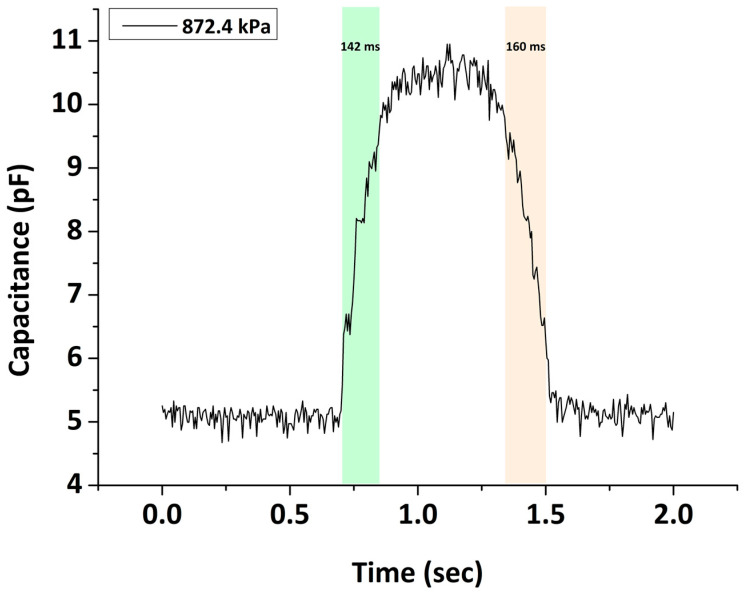
Sensor response and recovery times under an applied pressure of 872.4 kPa.

**Figure 11 sensors-22-09725-f011:**
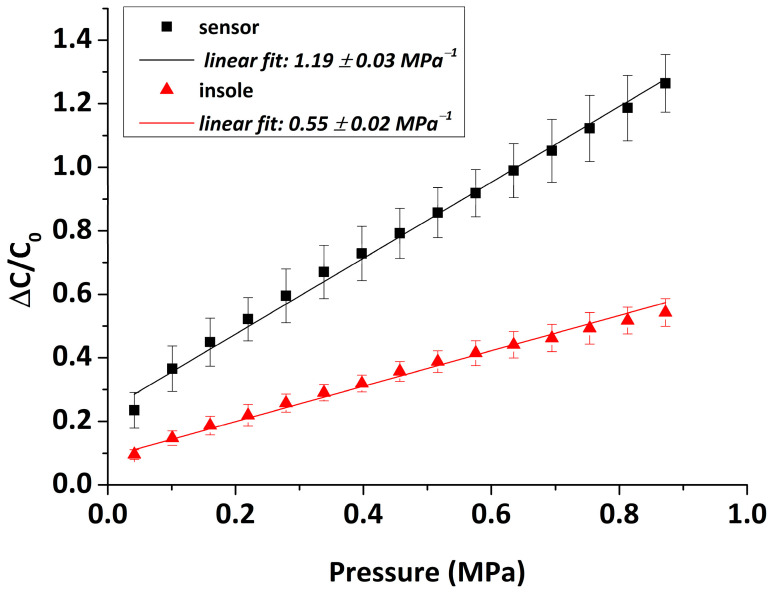
Average relative capacitance response of the 3D-printed capacitive pressure sensors as a function of the applied pressure measured outside (black) and inside the insole (red). The slopes correspond to the sensitivities.

**Figure 12 sensors-22-09725-f012:**
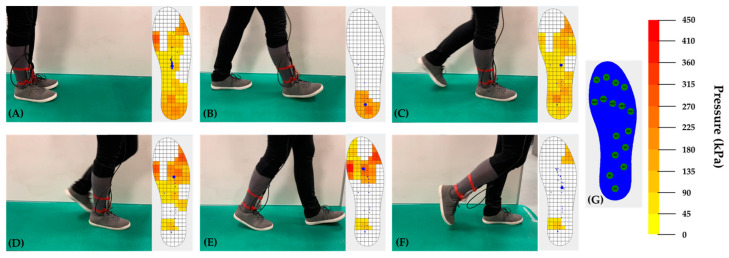
Plantar pressure distribution (heatmap) of the right foot under the main phases of a normal gait cycle: (**A**) stance, (**B**) heel strike, (**C**) foot flat, (**D**) midstance, (**E**) heel off and (**F**) toe off, (**G**) insole sensors arrangement and color map of the pressure range.

**Figure 13 sensors-22-09725-f013:**
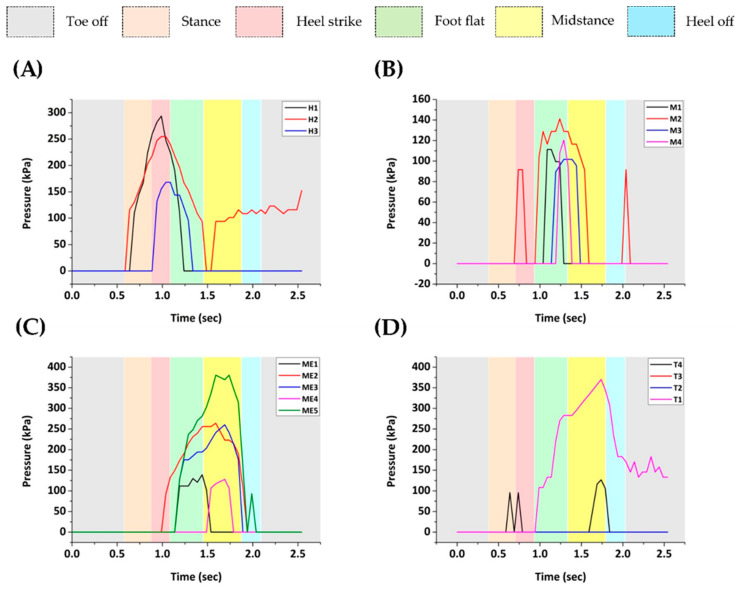
Real-time plantar pressure signals obtained from the right foot of the subject during a normal gait cycle using the developed smart insole system. Sensors monitored four areas: (**A**) calcaneus, (**B**) plantar arch, (**C**) metatarsals and (**D**) phalanges.

**Table 1 sensors-22-09725-t001:** Printing parameters applied for the 3D printing of the different materials.

3D Printing Settings	Protopasta Conductive PLA	Filaflex 70A/82A TPU
Infill (%)	100	
Extruder temperature (°C)	215	230
Bed temperature (°C)	45	
Print speed (mm/s)	15–20	
Layer height (mm)	0.1	
Nozzle diameter (mm)	0.6	
Flow rate (%)	110	
Flow width (mm)	0.6	
Retraction speed (mm/s)	60	

**Table 2 sensors-22-09725-t002:** Analysis of the cost per sole.

**3D Printing**			
**Part**	**Mass (gr)**	**Material**	**Cost (EUR)**
Insole case (top and bottom)	82	Filaflex 82A	4.57
Sensor—Conductive parts	4	Protopasta	0.50
Sensor—dielectric	4	Filaflex 70A	0.38
**Electronics**			
**Part**	**#**	**Characteristic**	**Cost (EUR)**
Cables	32	30 AWG	4.00
Microcontroller	1	Arduino mega	41.00
USB cable	1		2.00
**Other**			
**Part**			**Cost (EUR)**
Adhesive conductive glue			4.00
Adhesive glue			1.00
Insulation tape			0.50
Full insole (including cables)			14.95
Monitoring system			41.00
Total			57.95

## Data Availability

The study did not report any data.

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
