# Peer review of "A 3D-Printed Capacitive Smart Insole for Plantar Pressure Monitoring"

_sensors, 2022, doi:10.3390/s22249725_

Round 1

Reviewer 1 Report

The paper is well written and the experiments and analysis carried out with precision.  As you point out there are many wearable insoles available and you might point out more clearly the novelty and advantage of your solution.

Author Response

Response to Reviewer 1 Comments

Point 1: The paper is well written and the experiments and analysis carried out with precision.  As you point out there are many wearable insoles available and you might point out more clearly the novelty and advantage of your solution.

Response 1: We would like to thank the reviewer for her/ his time spent on this research work. We appreciate the positive feedback. We have pointed out more clearly the novelty and advantage of this work.

Reviewer 2 Report

The article is extremely well conceived and written. I don't want to invent any complaints, except for small typographical errors (SPACE in some places), everything is excellent.

Author Response

Response to Reviewer 2 Comments

Point 1: The article is extremely well conceived and written. I don't want to invent any complaints, except for small typographical errors (SPACE in some places), everything is excellent..

Response 1: We would like to thank the reviewer for the time she/ he spent on our research work. We appreciate a lot the positive feedback. We have gone through the paper multiple times to correct any typo and syntax errors.

Reviewer 3 Report

1.      This paper introduces a 3D-printed capacitive smart insole for plantar pressure monitoring. The paper provides good reference information and has great practical significance. The results are satisfactory.

2.      The paper is written well. These are some grammatical errors and typos. For example, in Line 78, "are relative inexpensive" should be "relatively"; in Fig. 5, in the title, "735.7 kPa" should be "753.7 kPa", etc. It is suggested for the authors to carefully proofread the paper before the next submission.

3.      Please provide some information on the cost to help the reader understand the feasibility of the product.

Author Response

Response to Reviewer 3 Comments

Point 1: This paper introduces a 3D-printed capacitive smart insole for plantar pressure monitoring. The paper provides good reference information and has great practical significance. The results are satisfactory.

Response 1: We would like to thank the reviewer for the time she/ he spent on our research work. We appreciate a lot the positive feedback.

Point 2: The paper is written well. These are some grammatical errors and typos. For example, in Line 78, "are relative inexpensive" should be "relatively"; in Fig. 5, in the title, "735.7 kPa" should be "753.7 kPa", etc. It is suggested for the authors to carefully proofread the paper before the next submission.

Response 2: We have gove through the paper multiple times and corrected any typo/ syntax errors.

Point 3: Please provide some information on the cost to help the reader understand the feasibility of the product.

Response 3: An analytical table (Table 2) at the conclusions section was added in order to explain the various costs, like the 3D printing and electronics parts that were used to develop this technology.

Reviewer 4 Report

In this paper (sensors-2045634), the 3D-printed capacitive smart insole for plantar pressure monitoring is presented. The results are good and the topic can attract a wide range of readerships. But there are some problems in the introduction, presentation, and discussion of the results. As such, some revisions are needed before possible publication. My specific comments are as follows:

1.      Introduction: The requirements of human gait monitoring pressure sensor (such as performance indicators) need to be discussed. The more recent research progress of pressure sensors for human gait monitoring needs to be reviewed and analyzed to highlight the difference and innovation of this work.

2.      The test equipment and conditions of the sensor should be given in detail.

3.      As a research paper, the sensing mechanism of the capacitive pressure sensor needs to be analyzed, may refer to J. Mater. Chem. C, 2021, 9, 13659–13667.

4.      The sensitivity is actually very low. Are there any advantages in performance compared with sensors in this field?

5.      Many references are out of date. It is suggested that references should be concentrated in the last three years, such as Sensors 2022, 22(21), 8327. In addition, check the reference format of the journal.

Author Response

Response to Reviewer 4 Comments

Point 1: Introduction: The requirements of human gait monitoring pressure sensor (such as performance indicators) need to be discussed. The more recent research progress of pressure sensors for human gait monitoring needs to be reviewed and analyzed to highlight the difference and innovation of this work.

Response 1: First of all we would like to thank the reviewer for her/ his time and all the frutfull comments that provided for improving the quality of this manuscript. The requirements of human gait monitoring pressure sensor were stated in the introduction. More recent research papers are reported in the introduction and we have highlighted the novelty on this work.

Point 2: The test equipment and conditions of the sensor should be given in detail.

Response 2: The test equipment is further analyzed and the environmental conditions of the sensor evaluation are mentioned. The 3D printing conditions were already mentioned in Table 1.

Point 3: As a research paper, the sensing mechanism of the capacitive pressure sensor needs to be analyzed, may refer to J. Mater. Chem. C, 2021, 9, 13659–13667.

Response 3: We provided the mathematical formula of the capacitance, which can be easily found through the basic principles of electromagnetism. However, we should state here that we are not using any mathematical formula for analysis, and this is the reason for not having it initially.

Point 4: The sensitivity is actually very low. Are there any advantages in performance compared with sensors in this field?

Response 4: The sensitivity of the designed 3D-printed capacitive pressure sensor is  with a working pressure range up to  (at least) while it drops to  when placed insole. This sensitivity should not be considered low when compared to other capacitive sensors for plantar pressure measurements [1]–[3] taking into consideration that for these type of applications the upper limit of pressure can be over  [4]–[6], hence a wide working pressure range is required.

References

[1]        S. W. Park, P. S. Das, and J. Y. Park, “Development of wearable and flexible insole type capacitive pressure sensor for continuous gait signal analysis,” Org. Electron., vol. 53, 2018.

[2]        Q. Zhang, Y. L. Wang, Y. Xia, X. Wu, T. V. Kirk, and X. D. Chen, “A low-cost and highly integrated sensing insole for plantar pressure measurement,” Sens. Bio-Sensing Res., vol. 26, 2019.

[3]        S. Yao and Y. Zhu, “Wearable multifunctional sensors using printed stretchable conductors made of silver nanowires,” Nanoscale, vol. 6, no. 4, 2014.

[4]        J. L. Chen et al., “Plantar Pressure-Based Insole Gait Monitoring Techniques for Diseases Monitoring and Analysis: A Review,” Advanced Materials Technologies, vol. 7, no. 1. 2022.

[5]        C. Giacomozzi, N. Keijsers, T. Pataky, and D. Rosenbaum, “International scientific Consensus on Medical plantar pressure measurement devices: Technical requirements and performance,” Ann. Ist. Super. Sanita, vol. 48, no. 3, 2012.

[6]        A. H. Abdul Razak, A. Zayegh, R. K. Begg, and Y. Wahab, “Foot plantar pressure measurement system: A review,” Sensors (Switzerland), vol. 12, no. 7. 2012.

Point 5 Many references are out of date. It is suggested that references should be concentrated in the last three years, such as Sensors 2022, 22(21), 8327. In addition, check the reference format of the journal.

Response 5: We replaced the out of date references and corrected the reference format.

Round 2

Reviewer 1 Report

This paper is clear and the work is well described.